# Synergistic Action of AMX Associated with 1,8-Cineole and Its Effect on the ESBL Enzymatic Resistance Mechanism

**DOI:** 10.3390/antibiotics11081002

**Published:** 2022-07-26

**Authors:** Ahmed Amin Akhmouch, Soukayna Hriouech, Aouatef Mzabi, Mariam Tanghort, Hanane Chefchaou, Adnane Remmal, Najat Chami

**Affiliations:** Department of Biology, Faculty of Science Dhar El Mahraz, University Sidi Mohammed Ben Abdellah, Atlas-Fez 1796, Morocco; ah.akhmouch@gmail.com (A.A.A.); hriouechsoukayna@gmail.com (S.H.); aouatef.mzabi@gmail.com (A.M.); mariamtanghort@gmail.com (M.T.); hanane.chefchaou@usmba.ac.ma (H.C.); chaminajat@gmail.com (N.C.)

**Keywords:** amoxicillin, 1,8-cineole, resistance, synergistic action, extended-spectrum beta-lactamases (ESBL), *Escherichia coli*, *Klebsiella pneumomniae*

## Abstract

The purpose of the present study is twofold. First, it aims to evaluate the synergistic action of the ß-lactam antibiotic; AMX is associated with 1,8-cineole on six clinical isolates of ESBL-producing *Escherichia coli* and *Klebsiella pneumoniae* strains. Second, it aims to determine the effect this association has on the ESBL enzymatic resistance mechanism. The synergistic action of AMX/1,8-cineole was evaluated using partial inhibitory concentrations (PIC), determined by a microplate, a checkerboard and time–kill assays. The effect of AMX/1,8-cineole associations on the ESBL enzymatic resistance mechanism was evaluated using a new optimized enzymatic assay. This assay was based on the determination of the AMX antibacterial activity when combined with 1,8-cineole (at subinhibitory concentrations) in the presence or absence of the ß-lactamase enzyme toward a sensitive *E. coli* strain. The results of both checkerboard and time–kill assays showed a strong synergistic action between AMX and 1,8-cineole. The results of the enzymatic assay showed that the combination of AMX with 1,8-cineole notably influences the enzymatic resistance of the reaction by decreasing the affinity of the β-lactam antibiotic, AMX, to the β-lactamase enzyme. All obtained results suggested that the AMX/1,8-cineole association could be employed in therapy to overcome bacterial resistance to AMX while reducing the prevalence of resistance.

## 1. Introduction

Antibiotic resistance has been a serious threat to public health in the past couple of decades, thus representing a challenge to medical research worldwide [1,2]. The production of ß-lactamase appears to be the most common resistance mechanism against ß-lactam antibiotics in Gram-negative bacteria and especially ESBL producers such as *E. coli* and *K. pneumoniae*, as confirmed by the World Health Organization [3]. To overcome resistance, many strategies have been considered. One of the most commonly adopted and successful strategies has been the co-administration of ß-lactam antibiotics with an enzyme inhibitor [4,5,6,7]. The inhibitors bind to ß-lactamase enzymes in an irreversible manner that deactivates the ß-lactamase, preventing it from destroying the accompanying ß-lactam antibiotic. The most commercially available inhibitor is clavulanate (CLA), which is used in combination with AMX. However, resistance to ß-lactamase inhibitors has also been reported [8]. As an alternative, several studies have reported the use of essential oils (EO) and their major compounds as synergistic agents to overcome ß-lactam antibiotics resistance [9,10,11,12]. In the present study, we have investigated the synergistic antibacterial effect of AMX when applied in combination with 1,8-cineole on multidrug-resistant bacteria. Then, the impact of this association on the ß-lactamase enzymatic reaction was examined.

## 2. Results

### 2.1. Determination of MIC and PIC Values

The MIC values of AMX and 1,8-cineole toward all of the six ESBL-producing strains were 50 mg/L and 100 mg/L, respectively. The obtained PIC values are shown in Table 1 and Table 2. IC50 and IC25 of AMX toward the six tested strains were 1.1–1.7 mg/L and 0.5–0.9 mg/L, respectively. IC40, IC30 and IC20 of 1,8-cineole toward the six tested strains were 6.2–7.1 mg/L, 2.3–3.1 mg/L and 0.4–1.3 mg/L, respectively.

### 2.2. The Checkerboard Assay

Antibacterial activity of the different associations of AMX/1,8-cineole at PIC towards the six ESBL-producing strains is shown in Table 3 and Table 4. The percentage of inhibition resulting when AMX at IC50 or IC25 was combined with 1,8-cineole at IC40, IC30 or IC20 was greater than the addition of the individual activities. Furthermore, the MIC of AMX and 1,8-cineole was significantly reduced in all bacteria. Total growth inhibition of all tested strains was observed when AMX at IC50 (1.1–1.7 mg/L) was associated with 1,8-cineole at IC40 (6.2–7.1 mg/L).

In terms of FIC index values of the different associations of AMX/1,8-cineole presented in Table 5, the combined application of AMX and 1,8-cineole produced a synergistic effect as defined by lower FIC indices. The FICI of AMX associated with 1,8-cineole was 0.08–0.10 against the six tested strains.

### 2.3. Time–kill Assay

Figure 1, Figure 2, Figure 3, Figure 4, Figure 5 and Figure 6 express the results of the time–kill assay of the six strains, when exposed to AMX and 1,8-cineole, alone or in combination, in MHB over 24 h. The application of 1,8-cineole or AMX alone (at their MICs) or in combination in all tested bacteria caused a significant reduction in bacterial counts (*p* < 0.05) in comparison with the control assay. No significant difference was found among the counts of all bacteria with AMX or 1,8-cineole alone (*p* > 0.05). However, 2-log reduction was obtained after 24 h of exposure when they were combined compared with either drug alone. Exposure of tested bacteria to the combination of AMX at IC50 and 1,8-cineole at IC40 resulted in a significant drop in the bacterial counts, and a 2-log reduction after only 4 h of exposure and a 4-log reduction after 8 h of exposure were obtained for all tested bacteria compared with the control. These results showed a significant inhibitory effect of AMX/1,8-cineole combinations on the cell viability of the tested bacteria.

### 2.4. Enzymatic Assay

The results of the calculated percentage inhibition of positive controls and experimental tube toward *E. coli*-sensitive strain are given in Table 6. Total growth inhibition was observed when AMX was applied alone at MIC. No growth inhibition was observed when 1,8-cineole was applied alone at sub-MIC and when the ß-lactamase was added to AMX at MIC. When adding both ß-lactamase and 1,8-cineole (at sub-MIC) to AMX (at MIC), the calculated percentage inhibition was 83.4% ± 1.1.

## 3. Discussion

The main aim of this study was, first, to evaluate the antibacterial activity of a ß-lactam antibiotic AMX associated with an EO major compound (1,8-cineole) towards clinical isolates of *E. coli* and *K. pneumoniae* ESBL-producing strains. All tested bacteria were resistant to the AMX. The MIC values obtained were 50 mg/L for AMX and 100 mg/L for 1,8-cineole. Associations between AMX and 1,8-cineole were made at partial inhibitory concentrations (IC 50% and IC 25% for AMX and IC 40%, IC 30% and IC 20% for 1,8-cineole). Total growth inhibition of all tested bacterial strains was obtained when AMX (at IC 50%) was associated with 1,8-cineole (at IC 40%). The MIC of associated AMX was around 25 times less than that of the single substance; the MIC of associated 1,8-cineole was about 15 times smaller than that of the single substance. FICI values were five times less than 0.5 for the association AMX/1,8-cineole. This synergy has been demonstrated in the time–kill experiments. Our results agree with previous literature data, which reported a high synergistic effect between Amoxicillin/clavulanic acid and 1,8-cineole on *Staphylococcus aureus* resistant to methicillin [13]. Currently, several studies reported synergistic interactions of essential oils such as *Thymus zygis* or *Thymus atlanticus* with antimicrobial agents including ampicillin, ciprofloxacin and vancomycin against *S. aureus*, *K. pneumoniae* and *E. coli* [14,15,16].

To our knowledge, the mechanism responsible for the synergetic action resulting from the combination of antibiotics and EO major compounds remains to be defined [17,18]. Nevertheless, recent studies have elucidated the mechanism of action of antibiotics associated with some essential oils against resistant bacteria [15,19].

Dumlupinar et al. [20] showed detailed cell damage of *K. pneumoniae* after treatment with a combination of *Pelargonium endlicherianum* essential oil with gentamicin and cefepime, by using scanning electron microscopy (SEM), resulting in increased efficacy of these antibiotics. In another study [19], the same authors proved that there was a synergistic effect of ampicillin associated with *Pelargonium endlicherianum* essential oil on *Neisseria meningitidis*. The results of this study illustrated that essential oils have two effects, the first, to increase membrane-permeability activity, and the second, to show that it had phagocytic activity in human leukocyte cells.

The most commonly known mechanism of resistance to ß-lactam antibiotics is the production of ß-lactamase [7,21], which is based on the hydrolysis of the ß-lactam ring inactivating the drug. ß-lactamase inhibitors such as clavulanic acid and sulbactam have been used to face antimicrobial resistance [22,23]. However, resistance to these ß-lactamase inhibitors has also been reported [24,25].

Although producing ESBL, the six tested strains were very sensitive to the AMX/1,8-cineole associations. This observation led us to the hypothesis that the presence of 1,8-cineole might influence the ß-lactamase enzymatic reaction. To verify this hypothesis, we have realized an enzymatic assay based on the simulation of the following ß-lactamase enzymatic reactions in presence of 1,8-cineole:Active ß-lactamase + AMX = Bacterial Growth
Inactive ß-lactamase + AMX = Growth Inhibition

As a bacterial growth indicator, the *E. coli* strain selected to be used in this assay was sensitive to ß-lactam antibiotics and did not produce any ß-lactamase enzyme. 1,8-cineole was used at sub-inhibitory concentrations to only observe the antibacterial activity of AMX. Therefore, no antibacterial activity of 1,8-cineole was observed.

Compared with AMX tested alone, in the presence of the ß-lactamase, no growth inhibition was obtained. These results show that ß-lactamase completely prevents the antibacterial activity of AMX. In contrast, when AMX is combined with 1,8-cineole, its antibacterial activity is protected in the presence of ß-lactamase. The results obtained were comparable to those obtained in the absence of the ß-lactamase enzyme. Knowing that 1,8-cineole was used at sub-MIC, the obtained antibacterial activity is due solely to AMX. This can be explained by the fact that the combination of AMX with 1,8-cineole clearly influences the enzymatic reaction by decreasing the affinity of AMX to ß-lactamase.

In conclusion, through the use of partial inhibitory concentrations (PIC), we have demonstrated a strong synergistic action when AMX is associated with 1,8-cineole tested at very minimal concentrations on the six clinical isolates of ESBL-producing bacterial strains. The results show that the antibacterial activity of AMX on ESBL-producing strains can be rehabilitated by associating AMX with 1,8-cineole. We obtained similar results when 1,8-cineole was replaced by carvacrol or eugenol EO major compounds (data not shown). Likewise, results demonstrate the synergistic effects when gentamicin or tetracycline are combined with α-pinene, thymol, eugenol or carvacrol against *S.aureus*, *E.coli* and *Aeruginosa* strains [26].

Secondly, we have demonstrated with the enzymatic assay that the association of AMX and 1,8-cineole protects the antibiotic by considerably decreasing ß-lactam affinity to ß-lactamase.

These results are supported by a spectroscopic study that demonstrates that the addition of 1,8-cineole allows the rearrangement of amoxicillin molecules in solution in the form of oligomers of 3–4 amoxicillin molecules [27]. Similar results were obtained with AMX/Clavulanic acid and ampicillin when associated with 1,8-cineole, Carvacrol or Eugenol EO major compounds (data not shown).

These findings support previous studies published in our patent in which we have demonstrated that combining AMX with EO major compounds reduces the probability of selection of resistant mutants [28].

This new concept was tested in vivo on animals and humans (clinical trials), and the results obtained (data not shown) open the path to a new drug currently commercialized by Sothema under the name Olipen (AMM: DMP/21/NNPdDMP/VHA/18). This product is used in therapy to overcome bacterial resistance to ß-lactam antibiotics while reducing the rate of resistance prevalence.

## 4. Materials and Methods

### 4.1. Culture Media

Mueller–Hinton agar (MHA) and Mueller–Hinton broth (MHB) (BIOKAR, Allonne France) were prepared and sterilized according to the manufacturers’ instructions.

### 4.2. Bacterial Strains

In this study, six ESBL-producing strains were used: three *E. coli* strains (P956, P933 and P7847) and three *K. pneumoniae* strains (H1878, H2001 and H1893). They were clinically isolated and obtained from the laboratory of bacteriology of the University Hospital Center (Centre Hospitalier Universitaire Hassan II—CHU) in Fez, Morocco. *E. coli* ATCC 8739 strain, sensitive to ß-lactam antibiotics, was used as a test of microorganism for the enzymatic assay. It was provided by the National Institute of Hygiene (INH, Rabat, Morocco).

The inoculum suspension was obtained by taking colonies from 24 h cultures on tryptic soy agar. These colonies were suspended in sterile saline (0.9% NaCl) and shaken for 15 s. The density was adjusted using previously established optical density/concentration standard curves. The suspensions were then diluted in MHB to reach the final concentrations (2 × 10^7^ CFU/mL for the six clinical isolates and 3.3 × 10^6^ CFU/mL for *E. coli* ATCC 8739).

### 4.3. Antibacterial Agents

Amoxicillin was purchased from Sigma Aldrich, Saint-Quentin-Fallavier, France (CAS Number: 26787-78-0) and diluted in sterile distilled water to obtain stock solutions (400 mg/L). EO major compound, 1,8-cineole, obtained from Sigma Aldrich (CAS Number: 470-82-6) was diluted in 0.2% (*v*/*v*) agar according to the method described by Remmal et al. [29].

### 4.4. ß-Lactamase Enzyme

The ß-lactamase enzyme was obtained from Sigma Aldrich (CAS Number: 9073-60-3). It was prepared according to the manufacturer’s instructions.

### 4.5. Determination of Partial Inhibitory Concentrations (PIC) of AMX and 1,8-Cineole Using a Microplate Assay Technique

The PIC of AMX and 1,8-cineole against the six bacterial strains were measured by the microplate assay [29,30] in MHB using sterile separate microplates (96 wells, 200 µL working well volume). Twelve serial dilutions of AMX (50–0.025 mg/L) and 1,8-cineole (100–0.05 mg/L) were prepared in sterile hemolysis tubes by successive dilutions 1/2. A measure of 100 µL of each dilution were introduced horizontally into twelve wells. Bacterial suspensions were then diluted in MH broth and plated in 96-well plates at a density of 2 × 10^7^ CFU/mL. Negative control was prepared by adding 200 µL of MHB to the twelve horizontal wells. Positive controls were prepared by adding 150 µL of MHB and 50 µL from bacterial suspension (2 × 10^7^ CFU/mL) to the twelve horizontal wells. Each concentration was performed in triplicate. The optical density was determined by a spectrophotometer (Versamax, Molecular Devices) at 540 nm at 0 h, and 22 h after incubation at 37 °C. The inhibition percentage of the six strains in the presence of each of the two antibacterial agents was calculated using the following formula [30]:% inhibition = [1 − ((ODT22 − ODT0)/(ODC22 − ODC0))] × 100

ODT0: OD of experimental well at 0 h.

ODT22: OD of experimental well after 22 h incubation.

ODC0: OD of positive control well at 0 h.

ODC22: OD of positive control well after 22 h incubation.

### 4.6. Checkerboard Assay

The determined PIC of AMX (IC50 and IC25) and 1,8-cineole (IC40, IC30 and IC20) were used to make different associations. The following associations of AMX and 1,8-cineole were prepared in sterile tubes:AMX at IC_50_ + 1,8-cineole at IC_40_
AMX at IC_50_ + 1,8-cineole at IC_30_
AMX at IC_50_ + 1,8-cineole at IC_20_
AMX at IC_25_ + 1,8-cineole at IC_40_
AMX at IC_25_ + 1,8-cineole at IC_30_
AMX at IC_25_ + 1,8-cineole at IC_20_

The antibacterial activity that resulted from these associations was evaluated using the microplate assay technique described previously. In a separate microplate for each strain, every association was prepared in triplicate by adding 50 µL of the prepared solution, 50 µL from suspension of microorganisms (2 × 10^7^ CFU/mL) and 100 µL of MHB. For each antibacterial agent, every PIC was prepared in triplicate. A negative control was prepared in triplicate by adding 200 µL of MHB to the twelve horizontal wells. A positive control was prepared in triplicate by adding 150 µL of MHB and 50 µL from the suspension of microorganisms (2 × 10^7^ CFU/mL) to the twelve horizontal wells. As described previously, the OD at 0 h and after 22 h of incubation at 37 °C was determined to calculate the inhibition percentage.

For each microorganism, the AMX/1,8-cineole associations were deemed synergistic, indifferent or antagonistic by calculating the fractional inhibitory concentration index (FICI) using the formula for each agent [13]:ΣFICI = FIC (𝐴) + FIC (𝐵).

FIC (𝐴) = (MIC (𝐴) in combination/MIC (𝐴) alone); FIC (𝐵) = (MIC (𝐵) in combination/MIC (𝐵) alone). The index values of the fractional inhibitory concentrations are interpreted as follows: FIC ≤ 0.5 = synergy; 0.5 < FIC ≤ 0.75 = partial synergy; 0.76 ≤ FIC ≤ 1.0 = additive; 1 < FIC ≤ 4 = no interaction (not differential); and FIC > 4 = antagonism.

### 4.7. Time–Kill Assay

The time–kill experiments were performed with selected antibacterial associations according to the results of the checkerboard assay. AMX and 1,8-cineole were tested alone and in association at MIC [31,32]. The mixtures were inoculated with an overnight culture of each of the six tested strains adjusted to a final density of 2 × 10^7^ CFU/mL. After 0, 2, 4, 6, 8 and 24 h of incubation at 37 °C, aliquots were withdrawn and diluted with physiological saline solution. The dilutions were spread onto MHA and the colonies were counted after incubation at 37 °C for 24 h. The number of colonies was expressed as log CFU/mL. Reduction in viable cell count ≥2 log_10_ after 24 h incubation in comparison with the cell count of the most active single substance was interpreted as synergy [31].

### 4.8. Enzyme Assay

The ß-lactamase assay was investigated to clarify whether 1,8-cineole had inhibitory activity on this enzyme. In preliminary studies, we determined the MIC of AMX and the sub-inhibitory concentration (Sub-MIC) of 1,8-cineole toward *E. coli*-sensitive strain using the microplate assay technique as already described above [29,30]. They were, respectively, 6 mg/L and 0.002 mg/L (data not shown). A suspension of 3.3 × 10^6^ CFU/mL was prepared from an overnight preculture of the tested strain grown in MHB at 37 °C. Positive controls and experimental tubes were prepared in triplicate according to Table 7. Using a spectrophotometer, the optical densities (OD) of each tube were determined at 540 nm at 0 h, and 22 h after incubation at 37 °C. The inhibition percentage of *E. coli* tested strain in each tube was then calculated as described previously [30].

### 4.9. Data Analysis

For each tested strain, all experiments were performed in triplicate. These data were analyzed with Student’s *t*-test and one-way ANOVA followed by Tukey’s multiple comparison test (Graph Pad Prism, version 5.03). Differences of *p* < 0.05 were considered statistically significant.

## 5. Conclusions

Our results show that the AMX/1,8-cineole association could be exploited in therapy to overcome bacterial resistance to AMX while reducing the rate of resistance prevalence.

## 6. Patents

Pharmaceutical Formulation Comprising Cineole and Amoxicillin (2019). https://patentscope.wipo.int/search/en/detail.jsf?docId=US250863563&docAn=16306262 (accessed on 1 June 2022).

## Figures and Tables

**Figure 1 antibiotics-11-01002-f001:**
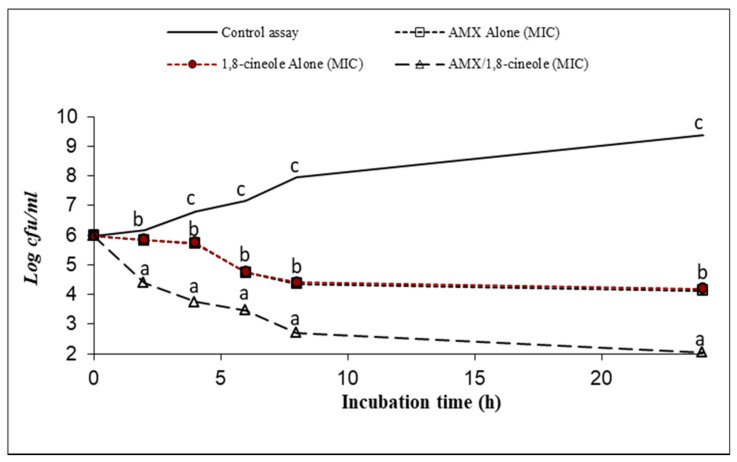
Time–kill curve of control assay, AMX alone at MIC (open squares), 1,8-cineole alone at MIC (filled circles) and AMX/1,8-cineole in combination at MIC (open triangles) against the *E. coli* P956 strain. The values that include different letters are significantly different from each other at *p* < 0.05.

**Figure 2 antibiotics-11-01002-f002:**
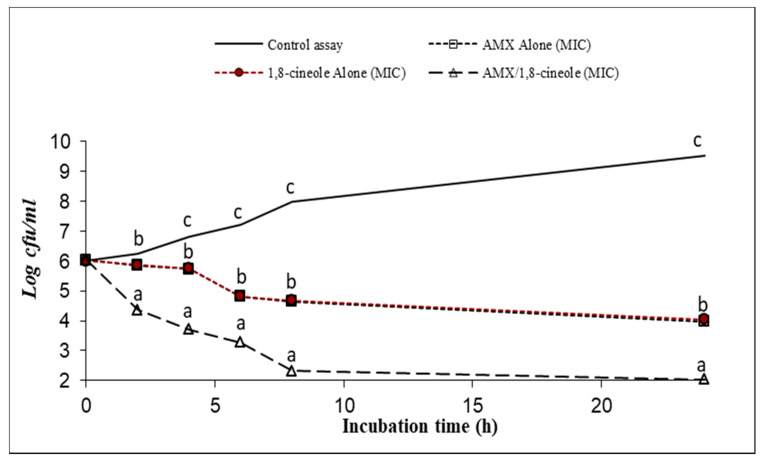
Time–kill curve of control assay, AMX alone at MIC (open squares), 1,8-cineole alone at MIC (filled circles) and AMX/1,8-cineole in combination at MIC (open triangles) against the *E. coli* P933 strain. The values that include different letters are significantly different from each other at *p* < 0.05.

**Figure 3 antibiotics-11-01002-f003:**
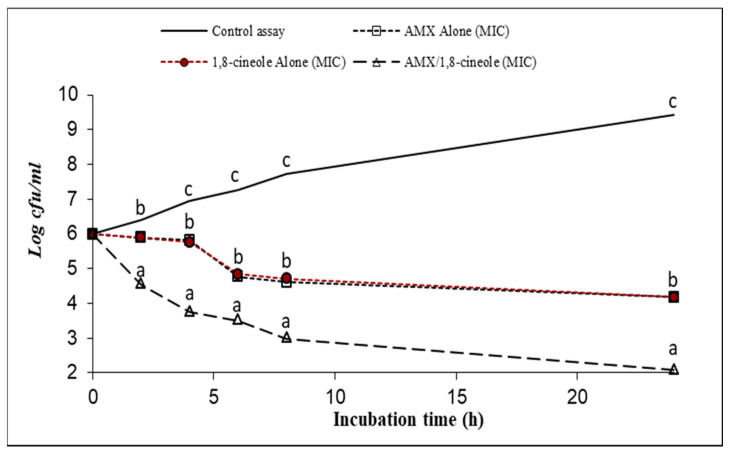
Time–kill curve of control assay, AMX alone at MIC (open squares), 1,8-cineole alone at MIC (filled circles) and AMX/1,8-cineole in combination at MIC (open triangles) against the *E. coli* P7847 strain. The values that include different letters are significantly different from each other at *p* < 0.05.

**Figure 4 antibiotics-11-01002-f004:**
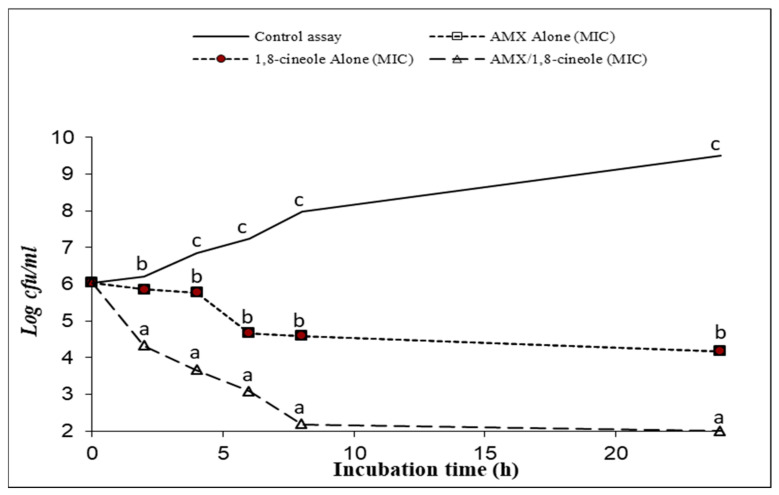
Time–kill curve of control assay, AMX alone at MIC (open squares), 1,8-cineole alone at MIC (filled circles) and AMX/1,8-cineole in combination at MIC (open triangles) against the *K. pneumoniae* H1878 strain. The values that include different letters are significantly different from each other at *p* < 0.05.

**Figure 5 antibiotics-11-01002-f005:**
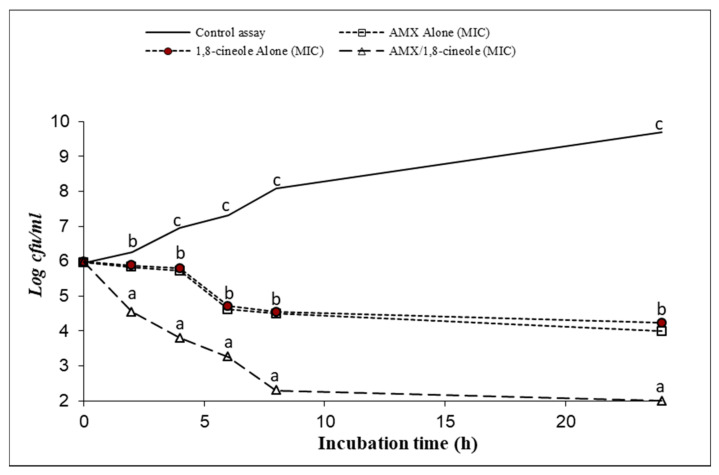
Time–kill curve of control assay, AMX alone at MIC (open squares), 1,8-cineole alone at MIC (filled circles) and AMX/1,8-cineole in combination at MIC (open triangles) against the *K. pneumoniae* H2001 strain. The values that include different letters are significantly different from each other at *p* < 0.05.

**Figure 6 antibiotics-11-01002-f006:**
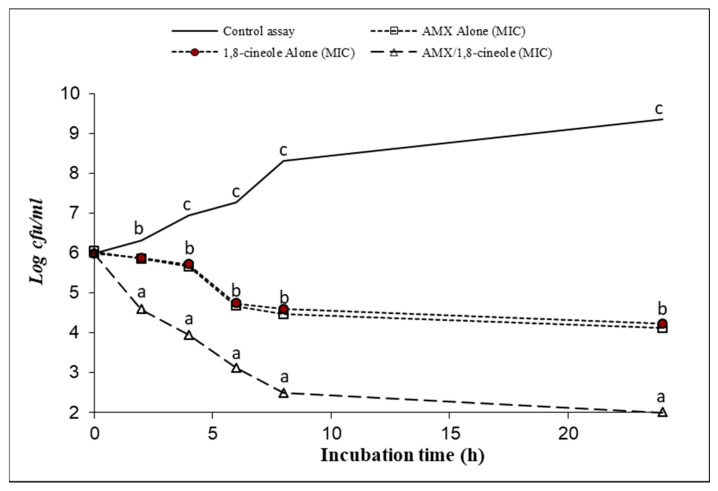
Time–kill curve of control assay, AMX alone at MIC (open squares), 1,8-cineole alone at MIC (filled circles) and AMX/1,8-cineole in combination at MIC (open triangles) against the *K. pneumoniae* H1893 strain. The values that include different letters are significantly different from each other at *p* < 0.05.

**Table 1 antibiotics-11-01002-t001:** PIC of AMX toward the six ESBL-producing strains.

Strains		IC_50_ (mg/L)	IC_25_ (mg/L)
* **E. coli** *	**P956**	**1.1 ± 0.25**	**0.7 ± 0.14**
	P933	1.3 ± 0.24	0.5 ± 0.10
	P7847	1.5 ± 0.13	0.8 ± 0.17
* **K. pneumoniae** *	H1878	1.6 ± 0.17	0.9 ± 0.09
	H2001	1.3 ± 0.12	0.6 ± 0.14
	H1893	1.7 ± 0.09	0.8 ± 0.12

**Table 2 antibiotics-11-01002-t002:** PIC of 1,8-cineole toward the six ESBL-producing strains.

Strains		IC_40_ (mg/L)	IC_30_ (mg/L)	IC_20_ (mg/L)
* **E. coli** *	P956	6.2 ± 0.20	2.6 ± 0.12	1.2 ± 0.16
	P933	6.8 ± 0.14	2.3 ± 0.09	1.3 ± 0.24
	P7847	6.5 ± 0.10	2.7 ± 0.12	1.2 ± 0.11
* **K. pneumoniae** *	H1878	7.1 ± 0.12	2.9 ± 0.12	0.4 ± 0.12
	H2001	6.9 ± 0.12	3.1 ± 0.10	0.7 ± 0.09
	H1893	6.8 ± 0.09	2.7 ± 0.09	0.6 ± 0.12

**Table 3 antibiotics-11-01002-t003:** Percentages of inhibition of AMX (IC 50%) associated with 1,8-cineole (IC 40%, IC 30% or IC 20%) toward the six ESBL-producing strains.

Strains		AMX/1,8-Cineole IC 50%/IC 40%	AMX/1,8-Cineole IC 50%/IC 30%	AMX/1,8-Cineole IC 50%/IC 20%
* **E. coli** *	P956	100.0 ± 0.0	92.5 ± 0.56	80.4 ± 0.69
	P933	100.0 ± 0.0	95.2 ± 0.75	81.5 ± 0.59
	P7847	100.0 ± 0.0	90.8 ± 1.29	79.4 ± 0.47
* **K. pneumoniae** *	H1878	100.0 ± 0.0	90.2 ± 0.68	82.8 ± 0.94
	H2001	100.0 ± 0.0	93.7 ± 0.80	81.2 ± 0.46
	H1893	100.0 ± 0.0	90.5 ± 0.25	85.2 ± 0.41

**Table 4 antibiotics-11-01002-t004:** Percentages of inhibition of AMX (IC 25%) associated with 1,8-cineole (IC 40%, IC 30% or IC 20%) toward the six ESBL-producing strains.

Strains		AMX/1,8-Cineole IC 25%/IC 40%	AMX/1,8-Cineole IC 25%/IC 30%	AMX/1,8-Cineole IC 25%/IC 20%
* **E. coli** *	P956	72.0 ± 0.38	62.1 ± 0.22	55.3 ± 0.75
	P933	73.1 ± 0.45	59.4 ± 0.32	51.2 ± 0.48
	P7847	70.1 ± 0.46	63.3 ± 0.36	53.4 ± 0.60
* **K. pneumoniae** *	H1878	74.4 ± 0.28	61.2 ± 0.26	56.3 ± 0.38
	H2001	72.4 ± 0.25	60.2 ± 0.77	53.9 ± 0.67
	H1893	71.1 ± 0.29	64.4 ± 0.34	52.6 ± 0.35

**Table 5 antibiotics-11-01002-t005:** FICI values for the different associations of AMX/1,8-cineole toward the six strains.

Strains		MIC Alone (mg/L)		MIC Combined (mg/L)		FIC-Index	Result
		AMX	1,8-Cineole	AMX	1,8-Cineole		
* **E. coli** *	P956	50	100	1.1	6.2	0.08	Synergy
	P933	50	100	1.3	6.8	0.09	Synergy
	P7847	50	100	1.5	6.5	0.10	Synergy
* **K. pneumoniae** *	H1878	50	100	1.6	7.1	0.10	Synergy
	H2001	50	100	1.3	6.9	0.10	Synergy
	H1893	50	100	1.7	6.8	0.10	Synergy

**Table 6 antibiotics-11-01002-t006:** Percent growth inhibition of *E. coli* strain in positive controls and experimental tube. The values that include different letters are significantly different from each other at *p* < 0.05.

		% Inhibition
**Positive controls**	**AMX**	100% ± 0.0 ^a^
	**1,8-cineole**	0% ± 0.0 ^c^
	**AMX + ß-lactamase**	0% ± 0.0 ^c^
**ß-lactamase + 1,8-cineole + AMX**		83.4% ± 1.1 ^b^

**Table 7 antibiotics-11-01002-t007:** Inocula, AMX, 1,8-cineole and ß-lactamase tubes content.

		MHB	Inocula ^1^	AMX ^2^	ß-lactamase ^3^	1,8-cineole ^4^
**Positive controls**	PC1	1000 µL	–	–	–	–
	PC2	970 µL	30 µL	–	–	–
	PC3	940 µL	30 µL	30 µL	–	–
	PC4	940 µL	30 µL	–	–	30 µL
	PC5	930 µL	30 µL	30 µL	10 µL	–
**Assay**		900 µL	30 µL	30 µL	10 µL	30 µL

^1^ 3.3 × 10^6^ CFU/mL of *E. coli* tested strain. ^2^ AMX was used at MIC (6 mg/L). ^3^ ß-lactamase was used at 0.03 U/mL. ^4^ 1,8-cineole was used at sub-MICs (0.002 mg/L).

## Data Availability

The data used to support the findings of this study are available from the corresponding author upon request.

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
