# Peer review of "Synergistic Action of AMX Associated with 1,8-Cineole and Its Effect on the ESBL Enzymatic Resistance Mechanism"

_antibiotics, 2022, doi:10.3390/antibiotics11081002_

Round 1

Reviewer 1 Report

The manuscript: Synergistic Action of AMX Associated to 1,8-Cineole and its Effect on the ESBL Enzymatic Resistance Mechanism by Ahmed Amin Akhmouch and co-authors concerns the important problem of searching for beta-lactamase inhibitors to both further use b-lactam antibiotics such as amoxicillin and also to limit the spread of gram-negative ESBL strains. 

Regarding methods - a few minor fixes:

line 222- correct the notation of CFU values

line 226 - provide the value of the stock solution of Amoxicillin

line 246-247 - correct the formula

line 277-278 - correct the formula

line 293 - the wording: "described previously" - specify the source of the citation or use the term "as already described above" if it concerns a description already given earlier in this work

Regarding results

1. The authors do not show the results from the plate tests that were the basis for selecting the AMX IC50 and IC25 concentrations as well as the 1,8-cineole IC40, IC30,  and IC20 concentrations. As was shown in the method, 12 serial dilutions of AMX and 1,8-cineole were tested. The MIC values for AMX and cineole for the six tested strains were also not shown. Such information is important because it constitutes the basis for the presented further results. These data must be shown, for example, in a supplement.

2. The authors also report that the percentage of growth inhibition of the tested strains against the combination of AMX and 1,8-cineole concentrations was higher than for individual substances alone. This is not apparent from the tables. The tables should be supplemented with the growth inhibition effect observed only for AXM and only for cineol.

3. The authors write that the MIC value of AMX and 1,8-cineole in combination was significantly reduced for all bacteria - this statement needs to be supported by data available in tables.  The baseline of MICs for AMX and 1,8-cineol is also not given in Tables 3 and 4. Tables must be supplemented.

4. Regarding table 5. - The analysis of the combination AMX/1,8 –cineol was obtained by calculating the fractional inhibitory concentration index (FICI). Table 5 should be supplemented with  MIC AMX in combination and alone as well as MIC 1,8-cineole in combination and alone.

Author Response

  • Response to Reviewer 1 Comments
  • Comment 1:  line 222- correct the notation of CFU values

Answer 1: The notation of CFU values was corrected. 3.3×106 CFU/mL. (Please see page:9, line:224)

  • Comment 2: line 226 - provide the value of the stock solution of Amoxicillin

Answer 2: We have added a short paragraph where we provide the value of the stock solution of Amoxicillin (Please see page:9, line:228).

Amoxicillin was purchased from Sigma Aldrich (CAS Number: 26787-78-0) and diluted in sterile distilled water to obtain stock solutions (400 mg/L). 

  • Comment 3: line 246-247 - correct the formula

Answer 3: The formula was corrected. (Please see page:9, line: 248-249)

                                   % Inhibition =

[1- (

ODT22-ODT0

)] X

100

ODC22-ODC0

  • Comment 4: line 277-278 - correct the formula

Answer 4: the formula was corrected. (Please see page:10, line:279-280)

FIC ≤ 0.5 = Synergy; 0.5 < FIC ≤ 0.75 = partial Synergy; 0.76 ≤ FIC ≤ 1.0 = Additive;

1 < FIC ≤ 4 = No interaction (not differential); and FIC > 4 = Antagonism.

  • Comment 5: line 293 - the wording: "described previously" - specify the source of the citation or use the term "as already described above" if it concerns a description already given earlier in this work.

Answer 5:

The wording: "described previously" was replaced in the manuscript by "as already described above" and the reference was also added. (Please see page: 10, line: 296)

Regarding results

  • Comment The authors do not show the results from the plate tests that were the basis for selecting the AMX IC50 and IC25 concentrations as well as the 1,8-cineole IC40, IC30, and IC20 concentrations. As was shown in the method, 12 serial dilutions of AMX and 1,8-cineole were tested. The MIC values for AMX and cineole for the six tested strains were also not shown. Such information is important because it constitutes the basis for the presented further results. These data must be shown, for example, in a supplement.

Table: Percentages of inhibition of AMX toward the six ESBL-producing strains

Strains

Concentrations (mg/L)

100

50

25

12.5

6.25

3.125

1.5625

0.78125

0.390625

0.195313

0.097656

0.048828

E. coli

P956

100

100

90±5.1

87±6.2

82.7±8.7

79.7±8.7

72.2±14.4

34.7±9.2

11.7±7.8

0.3±0.9

0

0

P933

100

100

90±2.5

84.2±1.8

80±1.3

73.2±1.5

58.7±16.5

39.5±4.2

28.3±9.5

13±4.7

0.3±0.9

0

P7847

100

100

89.8±2.8

85±2.5

79±3.8

74±3.3

49.5±4.2

32.7±9.4

13.8±12.5

4.2±6.4

0

0

K. pneumoniae

H1878

100

100

89.3 ±5.5

86±6.8

81.8±9

78.3±9.4

47.8±6.7

28.0±8.5

8±7.7

1±1.9

0

0

H2001

100

100

89.8±2.2

85±1.6

80.7±1.2

74.5±2.2

58.2±5.9

38.2±11.8

23.7±8.3

8.2±7.2

0

0

H1893

100

100

87.8±3.9

84.5±4

81.3±3.2

77±5.5

43.2±2.9

31.7±3.6

20±5.3

2.8±6.9

0

0

Table: Percentages of inhibition of 1,8-cineole toward the six ESBL-producing strains

Strains

Concentrations (mg/L)

100

50

25

12.5

6.25

3.125

1.5625

0.78125

0.390625

0.195313

0.097656

0.048828

E. coli

P956

100

81.7±2.5

53.9±1.9

43.2±1.8

39.7±1.1

36.9±1.4

29.3±5.5

17.8±4.2

0

0

0

0

P933

100

83±2.1

54.8±2.3

40.9±1.4

37.8±0.8

35.8±1.5

32.5±1.5

19.8±6

0

0

0

0

P7847

100

80.5±3

58.7±13.7

41.6±0.7

38±0.8

36.2±1.5

32.7±1.7

15.3±3.8

0

0

0

0

K. pneumoniae

H1878

100

76.5±3.2

51.7±1.4

41.5±1.1

37.2±0.7

33.9±1.8

31.4±1.7

18.5±3.2

0

0

0

0

H2001

100

78.7±3.5

53.3±2.5

42.4±1.5

38.1±2.5

34±1.3

20.1±4.8

9.4±5.5

0

0

0

0

H1893

100

82.3±2

51.2±1.3

43.2±2.2

38±2.0

34±1.4

29.9±5.4

18.3±5.4

7.2±5.0

0

0

0

The results of IC50 and IC25 for AMX, and IC20, IC30, and IC40 for 1,8- cineole were then inferred from their curves. 

  • Comment The authors also report that the percentage of growth inhibition of the tested strains against the combination of AMX and 1,8-cineole concentrations was higher than for individual substances alone. This is not apparent from the tables. The tables should be supplemented with the growth inhibition effect observed only for AMX and only for cineol.

Answer 2:

We included the MIC values ​​for AMX alone and 1,8-cineole alone in table 5. (Please see page: 3, line: 273).

  • Comment The authors write that the MIC value of AMX and 1,8-cineole in combination was significantly reduced for all bacteria - this statement needs to be supported by data available in tables.  The baseline of MICs for AMX and 1,8-cineole is also not given in Tables 3 and 4. Tables must be supplemented.

Answer 3: 

The MIC values ​​for AMX alone and 1,8-cineole alone were added in table 5. (Please see page: 3, line: 273).

  • Comment Regarding table 5. - The analysis of the combination AMX/1,8 –cineole was obtained by calculating the fractional inhibitory concentration index (FICI). Table 5 should be supplemented with MIC AMX in combination and alone as well as MIC 1,8-cineole in combination and alone.

Answer 4: 

The MIC values ​​for AMX alone and 1,8-cineole alone were included in table 5. (Please see page: 3, line: 273).

Strains

MIC alone (mg/L)

MIC Combined (mg/L)

FIC-Index

Result

AMX

1,8-Cineole

AMX

1,8-Cineole

E. coli

P956

50

100

1.1

6.2

0.08

Synergy

P933

50

100

1.3

6.8

0,09

Synergy

P7847

50

100

1.5

6.5

0,10

Synergy

K. pneumoniae

H1878

50

100

1.6

7.1

0,10

Synergy

H2001

50

100

1.3

6.9

0,10

Synergy

H1893

50

100

1.7

6.8

0,10

Synergy

Reviewer 2 Report

This is a standard study of antimicrobial activity. The authors used the common methods to asses antimicrobial activity of a beta-lactamase inhibitor of natural origin. It is generally a well-written article and it can be seen the authors are familiar with the testing methodology.

Line 80: Please clarify the phrase. Authors say there is a log2 reduction after 24h when AMX is combined with 1,8-cineole. However, at line 83 they say is a log4 reduction after 8 hours of exposure.

Author Response

  • Response to Reviewer 2 Comments
  • Comment 1: 

Line 80: Please clarify the phrase. Authors say there is a log2 reduction after 24h when AMX is combined with 1,8-cineole. However, at line 83 they say is a log4 reduction after 8 hours of exposure.

       Answer 1:  

We have added a short sentence stating that when we combined AMX with 1,8-cineole, there was a 2-log reduction after 24 hours of exposure compared to either drug alone, and a 4-log reduction after 8 hours of exposure was obtained for all tested bacteria compared to the control. (Please see page:3, line:81).

Round 2

Reviewer 1 Report

All comments of the reviewer were taken into account. The scientific value of the manuscript was strengthened.